# How Quantum Mechanics Requires Non-Additive Measures

**DOI:** 10.3390/e25121670

**Published:** 2023-12-18

**Authors:** Gabriele Carcassi, Christine A. Aidala

**Affiliations:** Physics Department, University of Michigan, Ann Arbor, MI 48109, USA; caidala@umich.edu

**Keywords:** quantum mechanics, measure theory, non-additive measures, information theory, statistical mechanics

## Abstract

Measure theory is used in physics, not just to capture classical probability, but also to quantify the number of states. In previous works, we found that state quantification plays a foundational role in classical mechanics, and, therefore, we set ourselves to construct the quantum equivalent of the Liouville measure. Unlike the classical counterpart, this quantized measure is non-additive and has a unitary lower bound (i.e., no set of states can have less than one state). Conversely, requiring that state quantification is finite for finite continuous regions and that each state counts as one already implies non-additivity, which in turn implies the failure of classical theory. In this article we show these preliminary results and outline a new line of inquiry that may provide a different insight into the foundations of quantum theory. Additionally, this new approach may prove to be useful to those interested in a quantized theory of space-time, as we believe this requires a quantized measure for the quantification of the independent degrees of freedom.

## 1. Introduction

It is well-established that standard measure theoretic tools cannot be used to characterize quantum probability as quantum theory is a non-classical (i.e., non-Kolmogorov) probability theory. To obviate this problem, various alternatives and extensions to measure theory and probability theory have been developed with different goals [1,2,3,4,5,6,7,8]. Our interest in measure theory, however, is not in its use to capture classical probability. Measures have another crucial use in physical theories: “counting”, or better “quantifying”, states. It is this use that will be the focus of this work.

As part of our overall project Assumptions of Physics [9], we have developed an approach, called reverse physics [10], which, starting from the laws, aims to find a minimal set of physical assumptions needed to rederive them.[note 1] We found that the ability to quantify states is a crucial component of classical theory, so much so that it is enough to recover the structure of phase space and the laws of evolution [16,17]. The former (i.e., the symplectic structure) is the only structure that allows one to quantify states and configurations over independent degrees of freedom in a way that is frame independent. If one imposes deterministic and reversible evolution, state quantification must be preserved, leading to Hamiltonian evolution (i.e., a symplectomorphism). All elements of classical mechanics, including canonical variables, the Lagrangian and the action principle [18], can be rederived and understood in terms of quantifying the flow of states. Given the formal similarity between classical and quantum mechanics, we believe a similar approach is possible for the latter. This raises the question: what is the quantum analogue of the Liouville measure, the measure that quantifies states in the classical theory?

Statistical mechanics and information theory mandate a link between entropy and state quantification, as the Shannon/Gibbs entropy over uniform distributions corresponds to the logarithm of the count of states. In the quantum case, we can start with the von Neumann entropy and define a measure that has the same connection. What we find is that this measure is non-additive and that its features can help us better understand what quantization is, why it requires both contextuality and the failure of classical probability, and why the latter can be recovered separately within each context. Therefore, we believe studying this object is useful not only for the above-stated goals of reverse physics, but more in general to get a better understanding of the foundations of quantum mechanics.

The aim of this article is to propose and articulate the goals of this new line of research. It provides the overall picture in which future works will add further mathematical detail and physical insights. In Section 2, we will show how measure theory is used to quantify elements in a set and how this is needed to assign probability and probability densities to each element. In Section 3, we will see how statistical mechanics and information theory mandate a link between entropy and state quantification, which we will use to define the quantum analogue of the Liouville measure. We will find that this measure is not additive.[note 2] In Section 4, we will see that this non-additivity is linked to non-contextuality. In Section 5, we will see that quantization means putting a unitary lower bound on the quantification of states, while keeping a finite value over finite regions. We will see that this very directly requires non-additivity and therefore non-contextuality. In Section 6, we will argue that quantization of space-time means using a quantized measure on the quantification of degrees of freedom. Lastly, in Section 7, we will outline a series of goals for a research program that aims to fully characterize the quantized measure.

## 2. Measure Theory and Quantifying States

As we said in the introduction, measure theory is not only used to characterize Kolmogorovian probability. More generally, measures are used to define the size of sets. Even in probability theory, since probability measures assign probability to sets of points, in order to associate probability to individual points, one needs two measures: the **probability measure** $p:\Sigma_{X}\to R$ and another measure quantifying the extent of the set, which we call **quantifying measure** $\mu :\Sigma_{X}\to R$.[note 3] Since we are interested in physical theories, the measure μ quantifies the number of states. Given a set *U*, the ratio $\frac{p(U)}{\mu (U)}$ gives us the average probability per unit of states. As we shrink *U* to a single point *x*, we will find the probability associated to that point, and how this is accomplished depends on the nature of μ.

In the discrete case, the quantifying measure is simply the counting measure: the cardinality of the set. Therefore, for a single point we simply have $\frac{p(\{x\})}{\mu (\{x\})}=\frac{p(\{x\})}{1}=p\left(x\right)$, which corresponds to the probability associated to the state *x*. The counting measure is fully determined by two properties: countable additivity, a defining property of measures, and that each state counts as one.

The continuous case is slightly different.[note 4] If *X* is a manifold, a set of local coordinates induce a Lebesgue measure, which can be used to quantify the number of possible value combinations. Note that a single state has measure zero, unlike the discrete case. The Lebesgue measure is defined by two properties: countable additivity and that the size of parallelepipeds corresponds to the volume expressed by the product of the differences of coordinates (i.e., $\mu \left(U\right)=\Delta x^{1}\Delta x^{2}\cdots \Delta x^{n}$). Naturally, this is a problem in physics given that the quantification of the number of states should be the same in all frames.

However, the structure of classical phase space (i.e., symplectic structure) is exactly the structure that allows one to define a quantifying measure for states that are invariant under change of frame [9,16,17]. The role of the quantifying measure in classical mechanics, then, is assigned to the Liouville measure, which we can understand as a Lebesgue measure on position and conjugate momentum (i.e., canonical coordinates).[note 5] As we shrink the size of the set, $\frac{p(U)}{\mu (U)}$ becomes the Radon–Nikodym derivative $\frac{dp}{d\mu }$, which gives us the probability density, not the probability itself.

Quantum mechanics, on the other hand, does not provide a clear measure for quantifying states.[note 6] When we characterize preparations and measurement outcomes, we may use the discrete or continuous quantifying measures in different cases for the same system. For example, consider a particle, if we are interested in a discrete observable, like the energy level of a harmonic oscillator, we would use the counting measure; if we are interested in a continuous observable, like position, we would use the Lebesgue measure associated to the variable. As for preparations, the same maximally mixed state for a qubit can be written as a discrete mixture $\frac{1}{2}\left|0\rangle \langle 0\right|+\frac{1}{2}\left|1\rangle \langle 1\right|$ or as a continuous uniform mixture over all possible states $\int \frac{1}{4\pi }|\psi \rangle \langle \psi |d\Omega$.

To be precise, the quantifying measures we considered really pertain to preparations and measurements, not the states themselves. It was the discrete or continuous character of the space of preparations that selected the appropriate space with the appropriate measure. But even if we have different preparation and measurement spaces, the state space of the system should always be the same, with a single way to count states. This, in other words, highlights the problem of quantum probabilities: while the spaces of preparations and measurement outcomes are both classical, they cannot be connected by classical means. That is, we cannot simply put a classical probability of transition between the two because we would be connecting qualitatively different quantifying measures.

To sum up, the typical foundational question is: how do we generalize probability measures for quantum mechanics? We are posing a different question: given that state-quantifying measures play an equally fundamental role, how do we quantify the number of states in quantum mechanics?

## 3. Entropy and State Quantification

While the standard structure of quantum mechanics does not give us an answer to our question, statistical mechanics and quantum information theory have an implicit answer. Statistical mechanics tells us that the thermodynamic entropy of a system, which is a physically meaningful quantity that has numerous empirical consequences, corresponds to the Shannon entropy for a classical discrete system, the Gibbs entropy[note 7] for a classical continuous system, and the von Neumann entropy for a quantum system.

There is another expression for entropy, often referred to as the fundamental postulate of statistical mechanics [21], which is the logarithm of the count of microstates: logμ(U).[note 8] In classical statistical mechanics, the measure is indeed the quantifying measure discussed above: the counting measure for the discrete case and the Liouville volume in the continuous case. The two expressions for entropy are linked as the Shannon/Gibbs entropy over a uniform distribution matches the logarithm of the measure of the support. That is, if $\rho_{U}$ is a uniform distribution over *U*, then $S\left(\rho_{U}\right)=\log\mu \left(U\right)$. In quantum mechanics, as we said before, we do not have a measure to quantify states. There is, however, a somewhat related expression: the maximum entropy attainable by a quantum system corresponding to a Hilbert space H is the logarithm of the dimensionality $\dim_{C}\left(H\right)$ of the space. Therefore, $S\left(\rho_{U}\right)\leq \log\left(\dim_{C}\left(\mathrm{span}\left(U\right)\right)\right)$.

A similar connection can be established in information theory. The Shannon/Gibbs/von Neumann entropy quantifies the minimum number of bits required on average to send a message from a given source.[note 9] This, again, is a quantity that leads to empirical consequences. If we have *n* bits at our disposal, we have $2^{n}$ possible combinations, possible messages, that can be encoded. Therefore, the logarithm of the number of potential messages corresponds to the number of bits that can encode those messages. Physically, we can associate a different state to each message to be encoded. Therefore, logμ(U) corresponds to the number of bits. This will exactly correspond to the number of bits required by a source characterized by a uniform probability distribution over that number of symbols.

While the above discussion does not cover all technical details, the only point we want to make is that the link $S\left(\rho_{U}\right)=\log\mu \left(U\right)$ is fully justified by both statistical mechanics and information theory. It is not a capricious request, but rather a necessary requirement stemming not from just one, but from two disciplines that have empirical consequences. Therefore, if quantum mechanics does not directly provide us with a measure to quantify states, we can use what it does provide, the von Neumann entropy, to define and calculate said measure.

We can start by calculating the simplest case, which is when U={ψ} is a single state. In this case, we have S(|ψ〉〈ψ|)=0 and, therefore, $\mu \left(\left\{\psi \right\}\right)=2^{0}=1$. The measure returns one for each state as in the classical discrete case.

The next case is when U={ψ,ϕ} is the set of two states. A uniform distribution over two states will correspond to the mixed state
(1)$$\rho =\frac{1}{2}\left|\psi \rangle \langle \psi \right|+\frac{1}{2}|\phi \rangle \langle \phi |.$$
The entropy of said state will depend on the probability $p={\left|\langle \psi |\phi \rangle \right|}^{2}$ in the following way
(2)$$S\left(\rho \right)=-\frac{1+\sqrt{p}}{2}\log\frac{1+\sqrt{p}}{2}-\frac{1-\sqrt{p}}{2}\log\frac{1-\sqrt{p}}{2}.$$
If the states are orthogonal, then p=0, and S(ρ)=1, therefore, $\mu \left(U\right)=2^{1}=2$. In this case we have that 2=μ({ψ,ϕ})=μ({ψ})+μ({ϕ})=1+1, and, therefore, μ is additive. Moreover, if *U* is an orthonormal basis, the von Neumann entropy $S\left(U\right)=\log(\dim_{C}\left(\mathrm{span}\left(U\right)\right))=\log|U|$ is the logarithm of the cardinality of *U*. Therefore, the quantifying measure μ(U)=|U| is equal to the counting measure. But this is a special case that corresponds to the highest entropy reachable with a mixture. In all other cases the entropy will be lower, and, therefore, in general, μ({ψ,ϕ})≤μ({ψ})+μ({ϕ}). In other words, the measure μ is not additive.

To sum up, the physically motivated way to quantify the number of states in quantum mechanics is, in general, not additive. When counting states in quantum mechanics, 1+1≤2.

## 4. Non-Additivity and Contextuality

While unusual claims are common in quantum mechanics, we want to have a clear, physically tenable, reason as to why the quantifying measure for quantum states must be non-additive. In the context of a spin 1/2 system, for example, why is it that the set of pure states $\{|z^{+}\rangle ,|z^{-}\rangle \}$ counts as two states while the set of pure states $\{|z^{+}\rangle ,|x^{+}\rangle \}$ counts as less than two?

Since we saw that the quantifying measure is indeed additive for a set of orthogonal states, this should make us think about contextuality: the impossibility to assign outcomes to all potential measurements of a quantum system. We have, in fact, derived that it is impossible to define an additive quantifying measure on all potential states, but we can do so on the outcomes of one measurement. The ability/inability to define additive measures is exactly the difference between classical and non-classical probability.

While contextuality is often presented as a somewhat abstract, almost metaphysical, property of quantum mechanics, we have a rather more down-to-earth view. The choice of the measurement means a difference in physical process of either the preparation or measurement device. Choosing which direction of spin to prepare or measure means, at some point, orienting some polarizer, magnetic field or making an equivalent change to a physical device. Something is changed in the physical world, not just in the system, but around the system, in its environment. If Amanda claims that a particle is in the $|z^{+}\rangle$ state, then Boris is able to infer something about Amanda’s preparation/measuring device.[note 10] Therefore, we can understand the context simply as those conditions in the environment that are necessary for the system to be found in that particular state. A set of orthogonal states, then, has the property that they can be found in the same external conditions.

Given that we have, in general, a correlation between system and environment, it is natural to understand how contextuality leads to a non-additive measure. Two states, in fact, should be counted as two only if they lead to a change in the system alone. That is, **states should be quantified all-else-being-equal**. While two states from different contexts are indeed two distinct states, stating we transitioned from one to the other does not only give us information about the system, it tells us something about the environment as well. The degree of incompatibility of observables, then, ultimately measures the degree of change when realizing the context physically.

So we have a clear, physically motivated reason as to why contextuality is linked to a non-additive measure. States should be quantified at-all-else-being-equal, meaning that possible correlations with the environment have to be removed.

## 5. Non-Additivity and Quantization

While we have clarified what it means for the quantifying measure to be non-additive, and how this is tied to non-contextuality, we have not shown why it should be linked to quantization. In fact, what exactly is quantization?

Before trying to answer that question, let us make a direct comparison between the quantifying measures for the three cases: the counting measure $\mu_{d}$ for the classical discrete case, the Liouville measure $\mu_{c}$ for the classical continuous case and the newly defined quantized measure $\mu_{q}$ for the quantum case. This requires a space where all three can be defined, which fortunately exists: the surface of a sphere. Note that the comparison is only at the mathematical level, so the only requirement is to find a mathematical space that can be used to represent states in the three cases. The sphere is used in classical mechanics to define the orientation of an angular momentum in three dimensions; the manifold is symplectic and allows both a Liouville measure and a counting measure. The (Bloch) sphere is used in quantum mechanics to represent the states of a two-state system, which can also represent the orientation of a spin one-half particle, and, therefore, it also allows a quantized measure.

First, we want to compare the behavior on a single point. The counting measure returns one, and so does the quantum measure. The Liouville measure, however, returns zero. Second, we want to compare the behavior on a finite region, an open set. The Liouville measure returns the solid angle, which is always going to be finite as it must be less than or equal to 4π. The von Neumann entropy of a mixed state over a two-state system is between zero and one, and, therefore, the measure must be between one and two. However, given that an open set on a sphere has infinitely many points, the counting measure over a finite region will be infinite. Lastly, we want to compare the additivity. Both classical measures are additive, while the quantum measure is not.

Let us list the three properties we have mentioned:single states count as onefinite continuous regions correspond to finite state quantificationstate quantification is additive on disjoint regions.

These seem very reasonable properties to assume for a quantifying measure of states. **However, these three conditions are incompatible**. If each state counts as one, and the measure is additive, then a finite continuous region must be infinite as it contains infinite points. **Of the three conditions, we can only pick two**. This is, in fact, what each measure does. The counting measure rejects the second property. The Liouville (and Lebesgue) measure discards the first. The quantum measure sacrifices the third.

So, in light of this, what is quantization? If we have a system characterized by continuous quantities, such as position, the state space will necessarily be a continuous space and finite regions will have to correspond to finite information, as those correspond to the finite precision measurements we can make.[note 11] If the regions are large enough, the Liouville (or Lebesgue) measure will work. But as we shrink a region, we will encounter, given the additivity, regions that contain less than one state, which does not make sense. Note that measure less than one corresponds to negative thermodynamic entropy, which does not make sense either.

Quantization, then, is fixing the quantifying measure so that it is bounded from below by one (and the entropy by zero): each state counts as one. However, this comes at a price: non-additivity. This non-additivity is ultimately why classical probability has to fail as well, given that additivity is one of its axioms. It is also the same reason as why quantization implies contextuality: the non-additivity means that some states cannot be prepared at-all-else-being-equal.

In our view, this provides a good conceptual basis to understand the failure of classical mechanics, the need for the departure from classical ideas and the need for quantum theory. We believe this, by itself, is already sufficient reason to start a research program to explore these ideas fully. However, since most physicists do not seem to be interested in this type of “clean-up work”, we can easily argue that a full characterization of the quantized measure is critical for the development of future physical theories.

## 6. Implication for Space-Time Quantization

We have seen that quantization can be understood as putting a unitary lower bound on the state quantification: no set of states can have fewer than one state. If we were to quantize space-time, what is it that we are quantizing?

The most fundamental theories that are available to us are field theories. In a field theory, the state is described by the value of the fields at each point in space. At each point, the value of the field is independent from the others: the degrees of freedom form a continuum.[note 12] We can therefore pose the question: how do we quantify the degrees of freedom?

If we have a region of space and we double the volume, we can imagine we are also doubling the number of degrees of freedom. Therefore, we can construct a volume measure from the metric tensor, which would give us an invariant volume that quantifies degrees of freedom, much in the same way that the Liouville measure quantified states.

Since the measure is finite for finite regions and is additive, as we shrink the region, we will find regions with less than one degree of freedom. This does not make sense. As before, we need to put a unitary lower bound on the measure and our measure cannot be additive. We need a quantized measure. Quantizing space-time, then, means putting a quantized measure on the number of degrees of freedom.

What does it mean that two degrees of freedom count less than two? Before answering that question, let us consider time. We said that if we double the spatial volume, we double the number of degrees of freedom. Suppose we have a space-time region and we double the time, are we doubling the number of degrees of freedom? We are certainly doubling the support of the function we have to specify for the field configuration, but we would not consider that to constitute doubling the degrees of freedom. Intuitively, we just think of those as being the same degrees of freedom that are getting different values in time. This is because we expect the future and past values to be linked by some dynamical equations, so the future values would be linked, in some way, to the past values. That is, the past and future value of the same field at some point are not independent. This may give us an insight into the physical meaning of the non-additivity of quantification of the number of degrees of freedom.

The same field at two spatial points may count less than two degrees of freedom simply because their values are not independent. The values of the field at two distant points can be chosen arbitrarily; therefore, the measure would be additive at long range. But as the points get sufficiently close, this would not be the case. So, at short scale, we need a quantized measure to quantify independent degrees of freedom. Note that the lack of independence of the field values in close-by regions may mean that high frequencies of the fields are damped: the values have to become closer. A quantized measure of the degrees of freedom would effectively provide a cutoff. Therefore, we have a story that is very well motivated and provides a possible solution for high energy divergences in quantum field theories.

While the development of a quantum theory of gravity is not our primary interest, the mathematical tools required to characterize the non-additive quantized measure seem to be of a similar nature to the ones that may be used to quantize space-time.

## 7. Open Questions and Future Work

Having outlined the scope and motivation for this new approach, we now outline the main questions we think need to be addressed by this new research program.

Both the counting measure and the Lebesgue/Liouville measures are unique in some sense. For the counting measure, once we define the value for a single point, typically one, the whole measure is determined. Similarly, once we have defined a real variable, the Lebesgue measure is determined. The Liouville measure requires a bit more structure, but once the canonical coordinates are chosen, the measure is determined as well. Therefore, the first question is whether these quantized measures are also unique in some sense.

While we have found some literature on non-additive measures and set functions [5,20], together with modified theories of integration, these seem to concentrate on monotonic measures: measures that increase as the set increases. Interestingly, the quantized measure does not have this feature. Consider the set $\{|z^{+}\rangle ,|z^{-}\rangle \}$. An equal mixture will return the maximally mixed state which, as we said, would correspond to a measure equal to 2. Now consider the set $\{|z^{+}\rangle ,|z^{-}\rangle ,\left|x^{+}\rangle \right\}$. An equal mixture of them will not correspond to the maximally mixed state, and, therefore, the measure will be less than two. The set is bigger, but the measure is smaller. Therefore, there may be new mathematical ground to be explored as well as conceptual ground. The physical meaning of non-monotonicity is not yet clear to us. We know it appears only when adding states that can be expressed as a superposition of the original set. One approach[note 13] would be to study the set of points that can be added without change to the measure.

We may want to understand the necessary conditions for a quantifying measure. Some may be derived from the strict concavity of entropy, given the connection between the two. Furthermore, note that the pairwise relationships between the measure over two points, the entropy of the equal mixture of two states and the probability of transition between two states are all invertible. This means that the relationship between two states defines the geometry of the whole state space. Therefore, there must be further consistency requirements that can be identified.

Another issue that needs to be understood is how the quantized measure behaves in composite systems and across multiple degrees of freedom. In the classical case, the measure is defined on the product of the corresponding σ-algebras. Ideally, we would like to show that this simple procedure cannot support a quantized measure, and the σ-algebra of the tensor product must be taken.[note 14]

The first spaces we should study are the symplectic ones as these are the ones that allow for frame independent densities. If we want to extend to a quantization of space-time, however, Riemannian spaces need to be studied as well. However, we have found a connection between the two: if the symplectic form is expressed in position and velocity, instead of canonical coordinates, its position-velocity component corresponds to the metric tensor. The metric tensor can then be understood as defining volumes in position-velocity instead of position-momentum. This may allow us to transfer the quantized measure onto the tangent bundle and then, hopefully, to the space itself.

Stepping back from the mathematical problem, there are also more conceptual issues that need to be understood. For example, what are the operational requirements to be able to quantify the number of states? It may turn out that the ability to operationally define and manipulate a system already provides all that is needed to define how to quantify states.

Finally, we want to stress that, contrary to what happens in other programs, the physical motivations for the mathematical generalization are tightly driven by the physics. We are not introducing concepts that, like negative probability, have no clear direct physical meaning. The goal is the quantification of states.

## 8. Conclusions

We have seen that, in physics, measure theory is not only used to characterize classical probability but also to quantify states in state spaces. Ideally, such a measure would have the following three conditions:single states count as onefinite continuous regions correspond to finite state quantificationstate quantification is additive on disjoint regions.

However, these conditions are not compatible, and only two of the three can be chosen. The classical discrete case discards the second; the classical continuous case discards the first; the quantum case discards the third. Quantization, then, can be understood as putting a unitary lower bound on the measure that quantifies states.

The lack of additivity on the whole space makes it impossible to use classical probability on the state space as a whole. Physically, the issue is that states must be counted at-all-else-being-equal, a feature broken by contextuality. But if the states do belong to the same context (e.g., they are the output of the same measurement), additivity is recovered. This tells us very directly why quantization requires contextuality, and why classical probability works within the same context. The fact that single states and finite continuous regions both have finite measure also tells us why quantum mechanics mixes features of the classical discrete and continuous cases.

We believe that further study and characterization of the non-additive quantized measure will not only bring more clarity to the interplay among the above concepts, but it may be necessary for the development of future physical theories. Given that, in a field theory, the number of independent degrees of freedom is taken to be proportional to the spatial volume, quantization of space-time can be understood as the quantization of the measure that quantifies the independent degrees of freedom. The development of physically motivated mathematical tools may help in that regard.

## Data Availability

This study generated no data.

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
