# Peer review of "How Quantum Mechanics Requires Non-Additive Measures"

_entropy, 2023, doi:10.3390/e25121670_

Round 1

Reviewer 1 Report

Comments and Suggestions for Authors

I found this to be a highly interesting paper. I am familiar with the authors's work on re-examining the assumptions underlying the foundations of physics in the light of our modern understanding of the mathematics (and the physics). I personally think that this is a very important initiative. With the increasing sophistication of experimental methods, foundational questions are becoming ever more amenable to experimental investigation. Many of our assumptions underlying interpretations in physics are based on historical (often dogmatized) traditions. While consensus is important is science, it can sometimes become an impediment. Under my other hat as a physician, I have seen many practices which have been considered sacrosanct abandoned once proper scientific scrutiny has bene applied to them. That would appear to be the goal of this line of research - to question deeply the foundations of physics and to not merely accept what is commonly accepted merely because it is standard practice - without scrutiny. 

The issue of probability in quantum mechanics is fundamental, since it is essentially a statistical model. Much is made in the literature of the differences between classical and quantum mechanics as if there describe two different realities - which makes no sense since there is only one reality. This paper makes an important step forward in understanding the commonalities and differences between classical and quantum mechanics in so far as their probability structure is concerned. I find that the paper makes a very clear argument, well reasoned and accessible to a general audience. It may well be the case that some physicists will have already made up their minds on this subject (especially those in the shut up and calculate camp) but ideological intransigence is not a substitute for analytical thinking. Mathematicians might argue that some of what is said here is well known, but that does not imply that it is known to a physics audience while it should be. It is a somewhat esoteric subject in mathematics, and so unlikely to be on the radar of most physicists. But  believe that it should be - especially for any physicist working in foundations research.

I find no obvious errors of reasoning in the paper. I believe it to be scientifically sound and so I do not find a reason to support anything other than acceptance.

I do have one question, more a point of clarification - in speaking of states the authors use the term |x+>, which I presume refers to a mixed or superposition state. I do not think that this is made clear in the discussion.

Author Response

I found this to be a highly interesting paper. ... I believe it to be scientifically sound and so I do not find a reason to support anything other than acceptance.

We thank the referee for the kind words. We wholeheartedly agree with the spirit of the comments!

I do have one question, more a point of clarification - in speaking of states the authors use the term |x+>, which I presume refers to a mixed or superposition state. I do not think that this is made clear in the discussion.

|x+> refers to a pure state prepared in the positive x direction (implicitly assuming a Cartesian coordinate system). We changed the text to:

"why is it that the set of pure states $\{ |z^+\>, |z^-\>\}$ counts as two states while the set of pure states $\{ |z^+\>, |x^+\> \}$ counts as less than two?"

Reviewer 2 Report

Comments and Suggestions for Authors

In the theory of dynamical systems, systems are characterized by two
basic traits, their state space and their evolution laws. Among the
parameters specifying the state space, in turn, is its sheer size,
i.e., the quantity of states it comprises. In physics, it appears to
be well-defined. However, both in classical and in quantum mechanics,
the answer is in fact far from obvious. In particular, in quantum
mechanics, the way states should be counted is an essential property
that distinguishes it from all of classical physics.

The present manuscript is dedicated to precisely this question, how
and why a measure on the state space of quantum mechanics becomes a
constituent feature of the theory, focusing in particular on its
non-additivity. Pointing out the close relation of state-counting
measures to entropy and information, the authors show how plausible
conditions on such a measure for quantum mechanics are inconsistent
with additivity. All in all, I think this is a very inspiring paper
that provides easy access to the subject even for readers not familiar
with foundational questions of quantum mechanics. The authors achieve
presenting their arguments, in spite of their highly abstract
character, in a comprehensible manner that avoids overly technical
terminology and extensive formalism wherever possible. The paper
certainly deserves being published in Entropy.

Following the argumentation, a number of remarks and objections came
to my mind, which taken into account by the authors, might contribute
to the readability and persuasiveness of the paper:

– The authors present this work as part of a major research project
they call reverse physics, "which, starting from the laws, aims to
find a minimal set of physical assumptions needed to rederive them".
Trying to contextualize it, I immediately thought of axiomatization.
Indeed, axiomatizations aims at exactly that, reducing a body of
theorems to a minimal set that contains them within its logical
implications, without redundancy or contradiction. It has been
introduced for mathematical theories, but should quite as well apply
to sufficiently formalized physical theory. So, what's the
difference? It it lies in the working direction, i.e., here,
extracting axioms from a given set of laws (analogous to inverse
scattering), it would be interesting to whether there exist meta-
theorems in logics indicating the prospects of this process, for
example, by determining the number of required axioms as a kind of
dimension of the space of propositions spanned by the laws.

– Concerning the specific case of quantum mechanics, I wwonder whether
the authors are aware of other current projects which perhaps are very
distinct as to their conceptual framework, but very close in spirit
to the present work. In particular, there exist research activities
aiming at an axiomatization of quantum mechanics that includes basic
postulates on the information content or capacity of quantum systems
(for a selection, not quite up to date, see my refs. [1-5] below).
Mentioning these efforts would be instructive for readers and relevant
regarding originality.

– Including also mixed states appears consequent in view of the
objectives of the project. In quantum mechanics, however, mixed states
occur only in open systems, by masking external components of
the system which, taken into account, would render the total system
closed and thus in a pure state. They therefore implicitly involve
additional states, even a macroscopic environment if, e.g.,
measurements are involved. I am not sure whether an attempt to count
states makes sense if the very borders of the system are not well
defined.

– Evidently, the non-additivity of quantum state measures is
intimately related to the possibility of finite overlaps of quantum
states, which in turn is a consequence of how quantum mechanics
defines distances between Hilbert-space vectors. This indicates a
relation between metric and measure on (hyper)spheres, which could add
an important argument to the paper.

– The authors base part of their reasoning on a comparison of
classical with quantum measures in the particular case of spherical
state spaces. I doubt that this analogy is correct. In the classical
case, this space could represent, for instance, the orientation of an
angular momentum in three-dimensional space. In quantum mechanics, it
refers to the Hilbert space of a two-state system. However, these
spaces are not compatible. The abstract spherical Hilbert space of a
two-state system\ has nothing to do with angle coordinates three-
dimensional position space. Moreover, if mixed states come into play,
say represented by a Bloch sphere, they do not correspond to
probability densities on the surface, as the authors insinuate, but to
points inside the sphere, i.e., to Bloch vectors with a radius less
than one.

– The argument, put forward in section 5, that a quantum state measure
attributing finite counts to finite continuous regions must be bounded
from below, is evident. An immediate consequence is the concept of
Planck cells, i.e., regions in phase space corresponding to that lower
bound. This direct relationship to one of the basic traits of quantum
mechanics might render this reasoning even more convincing. A further
consequence, distinguishing quantum mechanics from a direct rigid
discretization of space and time, notably to limit the information
content of these quantities, is that quantization fixes the size, but
in no way the shape of Planck cells.

– An interesting historical aspect is that quantization as a
consequence of a boundedness of entropy has been anticipated clearly
by Planck, in his derivation of a finite quantum of action, and even
earlier by Gibbs, proposing his paradox, and by Boltzmann himself,
referring to discrete "complexions" in his definition of entropy.

– Sections 2 and 3 have the same title ("Measure theory and
quantifying states"). Certainly an unwanted mishap.

[1]  Anton Zeilinger, Found. Phys. 29, 631 (1999).
[2]  Lucien Hardy, arXiv:0101012v4 [quant-ph] (2001).
[3]  Rob Clifton, Jeffrey Bub, and Hans Halvorson, Found. Phys. 33,
     1561 (2003).
[4]  Jeffrey Bub, Found. Phys. 35, 541 (2005).
[5]  Giulio Chiribella, Giacomo Mauro D’Ariano, and Paolo Perinotti,
     Phys. Rev. A 84, 012311 (2011).

Author Response

We thank the reviewer for the positive and interesting feedback. We really appreciate they feel we were able to present a highly technical and abstract subject in a clear and accessible way. Many comments go beyond the scope of the present paper and would certainly invite interesting discussion. If the reviewer is inclined, after the review process is completed, we would be delighted to continue the conversation.

– The authors present this work as part of a major research project
they call reverse physics, "which, starting from the laws, aims to
find a minimal set of physical assumptions needed to rederive them".
Trying to contextualize it, I immediately thought of axiomatization.
Indeed, axiomatizations aims at exactly that, reducing a body of
theorems to a minimal set that contains them within its logical
implications, without redundancy or contradiction. It has been
introduced for mathematical theories, but should quite as well apply
to sufficiently formalized physical theory. So, what's the
difference? It it lies in the working direction, i.e., here,
extracting axioms from a given set of laws (analogous to inverse
scattering), it would be interesting to whether there exist meta-
theorems in logics indicating the prospects of this process, for
example, by determining the number of required axioms as a kind of
dimension of the space of propositions spanned by the laws.

It would probably take a full paper to delve into these issues. First, it should be clear that mere axiomatization does not provide much insight into a physical theory. After all, the postulates of quantum mechanics can be considered axioms of the theory. In our mind, the issue is that the starting points, whatever we call them, should ideally be understandable in clear physics terms, which presents some additional problems.

The first issue is the transition between the informal physical definitions (which are needed as they connect to experiments) to the formal language of the theory. During discussions, at least two philosophers of science argued that this was one of the key new elements in Galilean thinking: he knowingly studied idealized pendulums or falling bodies neglecting friction. Therefore there are considerations to be had even before the formal system is set up. We suspect, in fact, that some sort of idealization/simplification is a prerequisite to any formal physical theory. A clear understanding of how the transition from the informal side to the formal side works is part of our goal.

The second issue is that the same assumption can be implemented formally in different ways. For example, the assumptions of determinism and reversibility can be implemented by requiring equal number of states to be mapped forward and backward in time; information on the initial state to correspond to equal information on the final state; the uncertainty for a peaked distribution to be conserved during evolution and so on. Therefore, you have one physical assumption that maps to multiple mathematical translations (i.e. multiple axioms) which can be proven to be equivalent. This is like the axiom of choice being equivalent to Zorn's lemma or the well-ordering theorem. Studying equivalences between these starting points is more important to us than any particular choice of starting point.

It was actually another philosopher that told us that our approach was similar to how reverse mathematics operates, and therefore we adopted the name. Some of these issues are mentioned in our paper on reverse physics. Some not, as we have been gradually becoming more aware of them. Ideally, we would work with someone from the philosophy side to give a more clear conceptual framework to the reverse physics approach.

– Concerning the specific case of quantum mechanics, I wwonder whether
the authors are aware of other current projects which perhaps are very
distinct as to their conceptual framework, but very close in spirit
to the present work. In particular, there exist research activities
aiming at an axiomatization of quantum mechanics that includes basic
postulates on the information content or capacity of quantum systems
(for a selection, not quite up to date, see my refs. [1-5] below).
Mentioning these efforts would be instructive for readers and relevant
regarding originality.

We were aware of Hardy and Chiribella et al., with the second considered closer as they are not constrained to quantum mechanics. It is probably best to refer to the reverse physics paper, as this was written to give a sense of the difference between our approach and the others. We added a footnote:

"To see how the reverse physics approach compares and contrasts with respect to other works that aim to find physically motivated starting points, for example in the context of quantum theory,[11-15] we refer to [10]."

– Including also mixed states appears consequent in view of the
objectives of the project. In quantum mechanics, however, mixed states
occur only in open systems, by masking external components of
the system which, taken into account, would render the total system
closed and thus in a pure state. They therefore implicitly involve
additional states, even a macroscopic environment if, e.g.,
measurements are involved. I am not sure whether an attempt to count
states makes sense if the very borders of the system are not well
defined.

The reviewer here is touching on another point that we are also working on: what are the assumptions/requirements needed to be able to define and characterize a physical system? Defining what happens at the boundary is indeeded crucial, and, in some sense, we believe that systems can only be defined if there are processes under which the system is, at least to some good approximation, isolated. These are details we will need to address in future works. We have added the following paragraph in the future work section:

"Stepping back from the mathematical problem, there are also more conceptual issues that need to be understood. For example, what are the operational requirements to be able to quantify the number of states? It may turn out that the ability to operationally define and manipulate a system already provides all that is needed to define how to quantify states."

That being said, we do believe that mixed states are a core concept. As for the objection raised, to our understanding, it is only improper mixtures (i.e. those describing a subsystem of an entangled system) that require another system. Proper mixtures (i.e. those that cannot be understood as coming from an entangled state) can arise in different ways, the most obvious being due to ignorance on the preparation. Moreover, if one can only perform experiments on a particular system, proper and improper mixtures are indistiguishable. It is only when we are able to define two independent systems that we can show that the mixture on one was improper. Therefore, when defining a single system, we can't technically make the distinction.

Additionally, all that we can prepare and measure in practice are mixtures. We can never perfectly prepare or measure pure states. Even if we prepare an electron with spin up, there will be an uncertainty in the direction. At any rate, these are issues that are important, but go outside the scope of this paper.

– Evidently, the non-additivity of quantum state measures is
intimately related to the possibility of finite overlaps of quantum
states, which in turn is a consequence of how quantum mechanics
defines distances between Hilbert-space vectors. This indicates a
relation between metric and measure on (hyper)spheres, which could add
an important argument to the paper.

The reviewer is again touching on another point we are already exploring. Both the spaces of classical and quantum mixtures are metric spaces under the Fisher-Rao metric. This can be understood as the infinitesimal version of the Jensen-Shannon divergence, which essentially tells us how the entropy increases under mixture. If two mixtures are non-overlapping, the JS divergence has maximum increase, both in classical and quantum mechanics. It should be possible to reframe this link in terms of state counting.

Given that it would bring a whole set of new concepts to the paper, which would take significant space and add complexity, we had decided not to include this insight. Though it is clearly a target for future work.

– The authors base part of their reasoning on a comparison of
classical with quantum measures in the particular case of spherical
state spaces. I doubt that this analogy is correct. In the classical
case, this space could represent, for instance, the orientation of an
angular momentum in three-dimensional space. In quantum mechanics, it
refers to the Hilbert space of a two-state system. However, these
spaces are not compatible. The abstract spherical Hilbert space of a
two-state system\ has nothing to do with angle coordinates three-
dimensional position space. Moreover, if mixed states come into play,
say represented by a Bloch sphere, they do not correspond to
probability densities on the surface, as the authors insinuate, but to
points inside the sphere, i.e., to Bloch vectors with a radius less
than one.

Note that we are looking only for a mathematical comparison of the measures. Clearly the state spaces are different and the spaces of mixtures are not comparable. However, note that the Bloch sphere is the state space for a spin one-half system, which is the orientation of an (intrinsic) angular momentum in three dimensional space. Therefore the comparison between the two is not that stretched. We added a paragraph to clarify:

"Note that the comparison is only at the mathematical level, so the only requirement is to find a mathematical space that can be used to represent states in the three cases. The sphere is used in classical mechanics to define the orientation of an angular momentum in three dimensions; the manifold is symplectic and allows both a Liouville measure and a counting measure. The (Bloch) sphere is used in quantum mechanics to represent the states of a two-state system, which can also represent the orientation of a spin one-half particle, and therefore it also allows a quantized measure."

– The argument, put forward in section 5, that a quantum state measure
attributing finite counts to finite continuous regions must be bounded
from below, is evident. An immediate consequence is the concept of
Planck cells, i.e., regions in phase space corresponding to that lower
bound. This direct relationship to one of the basic traits of quantum
mechanics might render this reasoning even more convincing. A further
consequence, distinguishing quantum mechanics from a direct rigid
discretization of space and time, notably to limit the information
content of these quantities, is that quantization fixes the size, but
in no way the shape of Planck cells.

The reviewer is again touching on one of our related research directions. In the reverse physics paper, we use the lower bound on entropy (which is the same as a lower bound on the state count) to motivate a classical analogue of the uncertainty principle. The issue of Planck cells, in fact, is that even knowing that there are cells already implies that one can define a boundary between them in a sharp way, but that definition would imply the ability to resolve the space between cells, which is a contradiction. We suspect that quantum mechanics is essentially the only way to implement a zero entropy lower bound in a consistent way. This would be equivalent to showing the uniqueness of the quantized measure. So, indeed, this is all related.

While, as the reviewer notes, there is a strong link between these ideas, we decided not to include them as we found that it may overwhelm some readers. One crazy idea at a time! :-)

– An interesting historical aspect is that quantization as a
consequence of a boundedness of entropy has been anticipated clearly
by Planck, in his derivation of a finite quantum of action, and even
earlier by Gibbs, proposing his paradox, and by Boltzmann himself,
referring to discrete "complexions" in his definition of entropy.

We were aware of some of Planck's work (a couple of philosophers made us aware of it) but we had no idea of similar efforts from Boltzmann. This is another topic of potential collaboration with someone more knowledgable with that literature: in a way, we are trying to understand how those arguments could be made to work to derive quantum mechanics.

– Sections 2 and 3 have the same title ("Measure theory and
quantifying states"). Certainly an unwanted mishap.

We are extremely grateful to the reviewer to have caught this! We modified the title of section 3 to "Entropy and state quantification".

Reviewer 3 Report

Comments and Suggestions for Authors

The authors suggest that some new kind of a non-additive measure to be used in quantum mechanics. At first glance, the idea looked quite promising to me. However, the more attentively I was reading the manuscript, the more disappointed I was. The authors failed to describe even the most basic features of heir construction. Which formulation of Quantum Mechanics do they have in mind? What is the role of set of points in this formulation? What do they call states and what do they call the number of states after all?

  To my mind, this manuscript is not suitable for publication.

Author Response

>The authors suggest that some new kind of a non-additive measure to be used in quantum mechanics. At first glance, the idea looked quite promising to me. However, the more attentively I was reading the manuscript, the more disappointed I was.

We are sorry the rest of the manuscript did not live up to the initial promise and expectations of the referee.

>Which formulation of Quantum Mechanics do they have in mind?

The standard formulation of Quantum Mechanics, for which pure states are rays of a Hilbert space (or equivalently points in the projective space) and mixed states are density matrices (i.e. positive semi-definite Hermitian operators with trace one. We added the following footnote:

"In this context, a pure state, or simply state, is mathematically represented by a ray of a Hilbert space (or equivalently a point in the projective space) and a mixed state is represented by a density matrix (i.e. a positive semi-definite Hermitian operator with trace one."

>What is the role of set of points in this formulation?

As mentioned in the text (line 62), in measure theory a measure assigns a value to a set of points. Our goal is to define a measure that given a set of (pure) quantum states (which are points in the state space), returns a value that quantifies the number of states.

>What do they call states and what do they call the number of states after all?

As mentioned above, states are simply the standard pure states in quantum mechanics. The number of states is given by the quantifying measure. In the classical discrete case this is given by the counting measure (i.e. the cardinality of the set) (line 69); in the classical continuous case this is given by the Liouville measure (i.e. volumes of phase space) (line 74); in the quantum case we defined it as the logarithm of the von Neumann entropy of a uniform mixture of those states (line 138).

Reviewer 4 Report

Comments and Suggestions for Authors

referee report enclosed as PDF

Author Response

It is often the case that we get reviews that altogether miss the point of what we are trying to do, and then block publication. Therefore we really appreciate this reviewer as they not only understood the point, but were also able to elaborate on it. We additionally appreciate the attitude of "letting people try new things."

More puzzlingly, however, the
function is not even monotonic. The authors do not, in my view, explain what
the meaning of that particular anomaly might mean. ...  I would
prefer to see the authors address the above remarks,

We did not explain the meaning of the non-monotonicity because, at this point, we have no clear physical understanding. That is also why we only mentioned it in the future work section. We have, at least, added a remark to acknowledge that this is the current state.

"Therefore there may be new mathematical ground to be explored as well as conceptual ground. The physical meaning of non-monotonicity is not yet clear to us. We know it appears only when adding states that can be expressed as a superposition of the original set. One approach\footnote{Also suggested by one anonymous reviewer.} would be to study the set of points that can be added without change to the measure."

Round 2

Reviewer 2 Report

Comments and Suggestions for Authors

The way the authors respond to my comments and complement their
answers with modifications of the manuscript is absolutely satisfactory. I am glad to conclude that in this case, even I as a
referee could learn quite a bit.

There are only three points where some doubts remain on my side:

– Measuring mixtures: If the present project aims at counting states
on basis exclusively of the density operator \rho, standard quantum
theory offers a simple measure, the inverse participation ratio (IPR),
defined as 1/tr[\rho^2}, which varies between 1 for a pure state and
the Hilbert space dimension for a maximally mixed state. Readers
familiar with this concept might appreciate a comparison. If, by
contrast, the system size accessible by measurements is the issue,
then it becomes an extremely diffuse issue, since the environment
inevitably comes into play, and its contribution depends on all kinds
of details of the coupling, the internal structure of the
environment, etc.

– Bloch sphere: I would insist that, if mixed states are included, the
Bloch sphere comprises the surface plus the interior, i.e., the
entire solid sphere, and thus does not compare well to the state space
of a classical angular of fixed magnitude in three dimensions.

– Planck cells: I am puzzled by the statement in the authors' reply that "The issue of Planck cells, in fact, is that even knowing that
there are cells already implies that one can define a boundary between
them in a sharp way ...". Such boundaries can certainly not be drawn.
Even in the simplest case of a coherent state, the exponentially
decaying shoulders do not allow for fixing a boundary. In the case of
classically chaotic systems, eigenstates are distributed approximately
evenly over the entire energy surface, with an extremely intricate
internal structure. The apparently paradoxical characteristic of
Planck cells is precisely that they cannot be outlined, yet their
fixed size delimits exactly the number of cells a given phase-space
volume can accommodate.

These comments are not meant as criticisms but just as spontaneous reactions to the authors' detailed reply. Otherwise, I now recommend
this paper unconditionally for publication in Entropy.

I noticed with delight that the authors dared chucking out the
otherwise inevitable Alice and Bob and replaced them with Amanda and
Boris. Why not go even further and employ Arnold and Berta, or even
Yolanda and Zoe?

Reviewer 3 Report

Comments and Suggestions for Authors

In the resubmitted version, the authors made some changes which imroved readability of the manuscript. However, in my view the proposal by the authors remains too vague. I cannot recommend publication of this manuscript.